# Loss of Phosphomannose Isomerase Impairs Growth, Perturbs Cell Wall Integrity, and Reduces Virulence of *Fusarium oxysporum* f. sp. *cubense* on Banana Plants

**DOI:** 10.3390/jof9040478

**Published:** 2023-04-16

**Authors:** Sayed Usman, Xinwei Ge, Yueqiang Xu, Qijian Qin, Jin Xie, Bin Wang, Cheng Jin, Wenxia Fang

**Affiliations:** 1College of Life Science and Technology, Guangxi University, Nanning 530004, China; 2Institute of Biological Sciences and Technology, Guangxi Academy of Sciences, Nanning 530007, China; 3State Key Laboratory of Mycology, Institute of Microbiology, Chinese Academy of Sciences, Beijing 100101, China

**Keywords:** *Fusarium oxysporum* f. sp. *cubense* tropical race 4, phosphomannose isomerase, virulence, cell wall, drug target

## Abstract

*Fusarium oxysporum* f. sp. *cubense* tropical race 4 (*Foc* TR4) causes Fusarium wilt of banana, necessitating urgent measures to control this disease. However, the molecular mechanisms underlying *Foc* TR4 virulence remain elusive. Phosphomannose isomerase is a key enzyme involved in the biosynthesis of GDP mannose, an important precursor of fungal cell walls. In this study, two phosphomannose isomerases were identified in the *Foc* TR4 genome, of which only *Focpmi1* was highly expressed throughout all developmental stages. Generated null mutants in *Foc* TR4 showed that only the Δ*Focpmi1* mutant required exogenous mannose for growth, indicating that *Focpmi1* is the key enzyme involved in GDP mannose biosynthesis. The *Focpmi1* deficient strain was unable to grow without exogenous mannose and exhibited impaired growth under stress conditions. The mutant had reduced chitin content in its cell wall, rendering it vulnerable to cell wall stresses. Transcriptomic analysis revealed up- and down-regulation of several genes involved in host cell wall degradation and physiological processes due to the loss of *Focpmi1*. Furthermore, *Focpmi1* was also found to be crucial for *Foc* TR4 infection and virulence, making it a potential antifungal target to address the threats posed by *Foc* TR4.

## 1. Introduction

*Fusarium* is a notorious fungal genus capable of producing harmful mycotoxins that cause plant and animal diseases, as well as mycotoxicoses in humans [1]. *Fusarium oxysporum* is a ubiquitous soil fungus consisting of more than 120 different formae speciales and is known for causing vascular wilting in a wide range of host plants, including vegetables, field crops, and plantation crops such as banana, cotton, tomato, cucumber, date palm, and oil palm. This vascular wilt disease results in vascular browning, leaf epinasty, stunted growth, wilting, and ultimately death of the host [2,3,4,5]. Aside from soil and infected plants, *F. oxysporum* has also been isolated from animals, where it causes life-threatening diseases in immunocompromised individuals [6]. Therefore, *F. oxysporum* fungal infections pose serious threats to human health and the global food supply, and the problem is only growing. Available antifungal drugs are limited, and emerging resistant pathogens are a growing concern, putting pressure on mycologists to find new antifungal targets and develop new strategies against fungal attacks.

Banana is one of the most important fruit crops worldwide [7,8]. However, *F. oxysporum* f. sp. *cubense* (*Foc*) is responsible for causing fusarium wilt of banana (Musa spp.) (FWB), leading to significant economic losses every year [9]. *Foc* has the ability to survive for extended periods (up to 20 years) in soil and can spread through contaminated plants, soil, tools, and water. Based on the host banana cultivars, four races of *Foc* have been identified, with *Foc* tropical race 4 (TR4) being the most disastrous, capable of attacking and infecting almost all banana cultivars. During the first half of the twentieth century, *Foc* TR1 infection completely destroyed the Gros Michel cultivar of banana, which was then replaced by the Cavendish cultivar. Later in the 1980s, *Foc* TR4 was discovered to infect the Cavendish cultivar. To date, *Foc* TR4 remains a serious threat as no efficient control strategy has been found to deal with FWB disease, due to the limited understanding of the underlying pathogenic mechanisms of *Foc* [10,11]. While a few virulence genes of FWB have been reported [12], the overreliance on chemical control of FWB has resulted in resistant pathogen populations and worsened the environment. To develop new strategies for FWB control, pathogenomics has emerged as a promising approach that enables the identification of putative genes involved in pathogenesis as potential targets. As there are no known resistant cultivars to *Foc* TR4 and efficient fungicides are not readily available, discovering new drug targets is critical to overcoming the threats posed by *Foc* TR4.

The fungal cell wall serves as an outermost layer of the cells that determines cell morphology and acts as a protective barrier against harsh environmental conditions, such as enzymatic attacks, freezing, stresses, osmotic changes, desiccation, and UV light [13]. It is composed mainly of glucan, chitin, and glycoproteins, with cell wall glycoproteins comprising 50–60% of the total cell wall mass and playing crucial roles in germination, cell wall and osmotic stresses, conidial separation, and virulence [14,15,16]. As the fungal cell wall is specific to fungi, distinct from plants’ cell walls, and absent in animal cells, it is considered an attractive target for developing novel and safe antifungal drugs [17,18]. Nucleotide sugars, such as UDP-glucose, UDP-GlcNAc, and GDP-Man, are the precursors for cell wall biosynthesis [19,20,21]. Previously, AGM1, UAP1, GNA1, PMM, PGM, and PGI have been reported as potential drug targets against *Aspergillus fumigatus* [19,20,21,22,23,24].

GDP-mannose (GDP-Man) is a crucial precursor for the biosynthesis of various molecules including glycolipids, galactomannan, glycosylphosphatidylinositol (GPI) anchors, and protein glycosylation, and is synthesized by three consecutive enzymes: phosphomannose isomerase (PMI), phosphomannose mutase (PMM), and GDP-mannose pyrophosphorylase (GMPP) [22]. PMI catalyzes the reversible reaction between fructose-6-phosphate (F6P) and mannose-6-phosphate (M6P) [25], and its functions have been reported in *Aspergillus flavus* [26] and *A. fumigatus* [27]. In *A. flavus*, PMI deficiency leads to defective cell wall and cell membrane organizations, impairment in growth and stress responses, and significantly reduced virulence in both plant and animal models [26]. Similarly, in *A. fumigatus*, deletion of *pmi* results in the inability to grow in the absence of exogenous mannose, abnormal morphology, cell wall defects, reduced conidiation, and abnormal germination polarity [27]. The functions of PMI have also been reported in several other pathogenic organisms, including *Cryptococcus neoformans*, *Leishmania mexicana*, and *Metharizium acridum* [28,29,30]. In comparison to the WT, the *C. neoformans* PMI disrupted mutant exhibits complete avirulence, poor capsule formation, and morphological abnormalities [30]. Similarly, in *L. mexicana*, the ∆*pmi* mutant exhibits attenuated virulence and a deficiency in glycoconjugate synthesis, although it can grow in the absence of exogenous mannose [28]. A study on the entomopathogenic fungus *M. acridum* showed that PMI deletion resulted in less virulence and more sensitivity to cell stress conditions [29]. These studies collectively demonstrate that PMI plays a central role in the development, cell wall synthesis, and virulence of various pathogenic organisms.

This study aimed to investigate the functional role of PMI in *Foc* TR4 by constructing a deficient mutant and conducting phenotypic analysis. The specific objectives of the study were to determine if PMI is involved in cell wall biosynthesis, energy production, and pathogenicity, as well as to assess its impact on growth, stress tolerance, maintenance of cell wall integrity, and pathogenesis of *Foc* TR4. The results of this study indicated that PMI plays a crucial role in all of these processes, suggesting that it may serve as a promising drug target for controlling FWB caused by *Foc* TR4.

## 2. Materials and Methods

### 2.1. Fungal Strains, Plasmids and Culture Conditions

*F. oxysporum* f. sp. *cubense* tropical Race 4 (*Foc* TR4) was used as the wild-type (WT) strain in the current study. *Foc* TR4 was cultured at 28 °C on PDA, MM, or in YPD liquid medium with shaking at 200 rpm as per experimental requirements. Conidia were washed with double distilled water (ddH_2_O) from 7 to 8 days culture on plates and counted using a hemocytometer. Mycelia were harvested by filtering two-day-old cultures through two layers of qualitative filter paper slowly. A modified pMD20-T vector and hygromycin selectable marker was used for the construction of mutants. All the vectors and plasmids were propagated in *Escherichia coli* DH5α.

### 2.2. Phylogenetic Analysis

The PMI sequence for *Foc* TR4 was identified via a tBlastn search of the genome using the *Saccharomyces cerevisiae* PMI sequence. PMI sequences from *F. oxysporum* f. sp. *cubense* (strain race 4), *F. oxysporum* f. sp. *pisi* HDV247, *F. oxysporum* f. sp. *vasinfectum*, *F. oxysporum* f. sp. *lycopersici* 4287, *F. oxysporum* f. sp. *rapae*, *F. oxysporum graminearum* PH-1, *Pyricularia oryzae* Y34, *Colletotrichum musicola*, *Verticillium dahlia* Vdls.17, *Pterula gracilis*, *F. oxysporum* f. sp. *cubense* race 1, and *Neurospora crassa* were downloaded from the National Center of Biotechnological Information (NCBI). A phylogenetic tree was generated using the neighbor joining method and bootstrap tests of 5000 replicates in the Mega 11 software [31].

### 2.3. Gene Expression Analysis of Focpmi1 and Focpmi2

For the analysis of *Focpmi1* and *Focpmi2* gene expression at the conidial, germination, and hyphae stages, total RNA was extracted from all three stages using the Transzol up plus RNA kit (transgen) according to the manufacturer’s instructions. Genomic DNA was removed, and cDNA was synthesized using the HiScript III RT SuperMix for qPCR (Vazyme).

A specific pair of primers for each gene was used for qRT-PCR analysis, and 200 ng of cDNA was used as the template. The qRT-PCR was conducted using the SYBR green qPCR master mix in a LightCycler 96 (Roche). The amplifications were carried out in a total volume of 20 µL, including 10 µL of SYBR green, 2 µL of cDNA template, 0.4 µL of each forward and reverse primer, and 7.2 µL of PCR-grade water. The reactions were performed using a two-step method: 95 °C for 30 s followed by 40 cycles of 95 °C for 10 s and 60 °C for 30 s.

To normalize the expression of both genes at different growth stages, the *FocEF1α* was used as an endogenous reference gene, and the 0 h stage was used as a control. The relative expression of each gene was calculated using the 2^−∆∆CT^ method [32]. The experiment was performed in triplicate and repeated three times to ensure accuracy and reproducibility. The resulting data were used to calculate the means and standard deviations. The primers used for qRT-PCR are listed in Appendix A.

### 2.4. Construction of Deletion Mutant and Revertant Strain

The ampicillin resistant modified pMD20-T vector and hygromycin (hygB) resistance marker was used for knockout construct preparation. A total of 1.1 Kb each of the upstream and downstream *Foc* TR4 *pmi* regions were amplified from the WT using primer pair P1 and P2, respectively. The hygromycin resistance marker gene was amplified from the pCSN44 vector using primer pair P3. The upstream region was inserted to *Asc* I-*Not* I, and the hygromycin part was incorporated into the *Not* I-*Fse* I site, while the downstream was inserted to *Fse* I-*Pac* I restriction site of the pMD20-T plasmid. Protoplasts were prepared according to previously described method [33]. Briefly, 10^8^ freshly harvested conidia from the WT were inoculated into 50 mL YPD liquid medium and cultivated at 28 °C for 5–7 h with shaking at 200 rpm. The germlings were harvested with a centrifuge at 5000 rpm (4 °C for 10 min) and washed twice with 0.7 M NaCl, then digested with an enzyme solution containing driselase (10 mg/mL) and lysing enzyme (10 mg/mL) in a 0.7 M NaCl solution at 30 °C for 50 min with shaking at 100 rpm. The protoplasts were collected by centrifugation at 5000 rpm (4 °C, 5 min), re-suspended in 0.7 M NaCl, and then placed in STC washing buffer (1.2 M sorbitol, 10 mM Tris-HCl pH8.0, 50 mM CaCl_2_). Finally, the protoplasts were diluted in STC buffer to a final concentration of 1 × 10^8^ protoplasts/mL. For transformation, 10 μg of PCR amplified knockout construct (*up-hygB-down*) in 100 μL STC buffer was mixed with 100 μL of protoplasts. Then, 50 μL 30% PEG8000 was added and incubated at room temperature for 20 min. After that, 2 mL of 30% PEG8000 was added and incubated for 5 min. Finally, 4 mL of STC buffer was added, mixed, and was then dissolved in molten regeneration medium containing 0.1% yeast extract (*w*/*v*), 0.1% Casein-Enzyme Hydrolysate (*w*/*v*), 0.8 M sucrose, and 1.5% Agar (*w*/*v*) at 50 °C and poured into petri dishes. The transformants were incubated at 28 °C for 12 h. Overlay regeneration medium with selective reagent hygromycin B (150 μg/mL) was added to the plates for further screening.

For gene complementation, the 4.3 kb PCR fragment containing upstream, the *pmi* gene, and downstream was amplified from WT genomic DNA using primer pair P8. Upstream and downstream regions were used as homologous regions and the PCR fragment was transformed into the protoplasts of the mutant. The reverted transformants were selected on plates without mannose supplementation.

### 2.5. Confirmation of the Mutant and Revertant Strains

The ∆*Focpmi1* and revertant (RT) strains were confirmed using both PCR and Southern blotting techniques. Genomic DNA from the WT, ∆*Focpmi1*, and RT strains was extracted using standard protocols. Four different primer pairs (P4, P5, P6, and P7) were used to validate the correct transformants. For Southern blot analysis, the genomic DNA of the WT, ∆*Focpmi1*, and RT strains were digested with *Hind*III, and the upstream fragment of the 1.1 kb non-coding region was used as a probe. The probe was labeled using the DIG-labeled hybridization protocol according to the manufacturer’s instructions (Roche).

### 2.6. Effect of Exogenous Mannose on the Growth of the Strains

To optimize the growth conditions of the ∆*Focpmi1* strain, YPD media containing different concentrations of mannose (0, 0.5, 2, 3, 5, 10, 15, 20, 30, and 50 mM) were prepared. Serially diluted conidia (10^6^–10^3^) of the WT, ∆*Focpmi1*, and RT strains were spotted on the plates and incubated at 28 °C for 48 h to evaluate the effects of exogenous mannose on the growth of the strains.

### 2.7. Colony Growth Rate and Conidiation Analysis

Fresh conidia from the WT, ∆*Focpmi1*, and RT strains were point inoculated on YPD plates, with 5 mM mannose supplementation for the mutant. After initial inoculation, the colony diameter was recorded for each strain along the same line continuously for 10 days. The colony morphologies of each strain were photographed on the 5th and 10th day of inoculation. For conidiation analysis, conidia from each plate were gently scraped using 0.2% Tween-20 and counted using a hemocytometer.

### 2.8. Sensitivity to Different Carbon Sources and Stresses

The growth of the mutant was assessed on different sole carbon sources. YPD or MM plates supplemented with various concentrations of fructose (0.5, 3, 5, 10, 15, 20, and 50 mM) were prepared. Serially diluted conidia (10^6^–10^3^) from all three strains were inoculated on the plates and incubated at 28 °C. Growth was also evaluated in the presence of other carbon sources, such as glucose (Glc), mannose, glucosamine (GlcN), glycerol, xylose, ethanol, arabinose, N-acetyl glucosamine (GlcNAc), sucrose, galactose, and maltose, all in MM media.

The sensitivities of the WT and ∆*Focpmi1* mutant to cell wall stressors (CR and CFW), oxidative stresses (sorbitol and H_2_O_2_), and ER stresses (TCEP and Brefeldin A) were assessed after point inoculation of serially diluted conidia (10^6^–10^3^) in YPD or MM supplemented with 5 mM mannose (referred to as MMM) at 28 °C.

### 2.9. Cell Wall Components Analysis

The cell wall components were determined using the previously described method [34]. Briefly, 2 × 10^8^ conidia of the WT, Δ*Focpmi1*, and RT strains were inoculated into MMM liquid medium and cultured at 28 °C at 150 rpm. After 72 h, mycelia were harvested via filtration and ground into a fine powder using liquid nitrogen. Next, 30 mL of SDS-BME solution (50 mM Tris, 50 mM EDTA, 2% SDS, 1 mM TCEP) was added to the mycelia powder and boiled at 100 °C for 20 min. After centrifugation at 8000 rpm for 10 min, the supernatant was discarded and the cell wall fractions were washed thoroughly six times with milliQ water and freeze-dried for three days. To determine the cell wall components, 10 mg of dry cell mass was mixed with 75 µL of 72% H_2_SO_4_ and left at room temperature for 3 h. The pellet was subsequently re-suspended in 0.95 mL of milliQ water and boiled for 4 h at 100 °C. Following this, the sample was neutralized to a pH of 7 using Ba(OH)_2_ and left overnight at 4 °C. The monosaccharide content of the supernatant was analyzed using HPAEC-PAD with a CarboPac PA-10 anion exchange column and an Amino trap guard column at room temperature. Elution was performed at room temperature using 18 mM NaOH with a flow rate of 1 mL/min.

### 2.10. Comparative Transcriptomic Analysis 

Fresh 100 µL conidia (1 × 10^8^/mL) from the WT and ∆*Focpmi1* strains were cultured in YPD medium, supplemented with 2 mM mannose for the mutant. The cultures were incubated at 28 °C at 200 rpm for 48 h. After 2 days of incubation, mycelia were filtered, washed twice with autoclaved water, dried well, and freeze-stored in liquid nitrogen. RNA isolation and sequencing were conducted by Biomarker Technologies (China). High-quality RNA samples were prepared and sequencing was performed on the Illumina sequencing platform. The differentially expressed genes (DEGs) were selected using DESeq2 with Fold Change ≥ 5 and FDR < 0.01. DEGs were visualized using a volcano plot and COG analysis was performed to characterize the biological functions of the DEGs and the metabolic pathways in which they are involved. The RNA-Seq Illumina reads were deposited in the National Center for Biotechnology Information Short Read Archive (NCBI-SRA) and are publicly available under the accession number PRJNA940601.

### 2.11. Virulence Assay

Banana seedlings were cultivated in plastic pots in a sterilized soil environment in a glass house. Banana plants of uniform size with an initial height of 6 cm were selected, removed from the soil, and had their roots washed with distilled water. The plants were then transferred to hydroponic pots containing the necessary nutrient solution. The plants were divided into four groups based on the strain treatment, namely CK, WT, Δ*Focpmi1*, and RT. For each strain, 12 plantlets were chosen as replicates, and the pathogenicity test was replicated three times.

The virulence assay was performed as previously described [35]. Briefly, fresh conidia of all three strains were harvested from 7-day-old liquid cultures at a concentration of 2 × 10^6^/mL. The roots of the banana seedlings were then completely immersed in nutrient solution containing conidia of the indicated strains in 50 mL falcon tubes, which were incubated at 28 °C. The growth rate was recorded, and the nutrient solution was supplied daily. After 28 days of initial inoculation, the banana plants were removed from the falcon tubes, and the final plant height and disease symptoms, such as yellowing of leaves and browning of rhizomes, were assessed. Disease severity was recorded and graded from 0 to IV for leaf yellowing and 0 to V for rhizome browning for each plantlet.

For estimation of fungal biomass, the corm tissues from each of the four treatments (CK, WT, Δ*Focpmi1*, and RT) were separately ground and weighed. The bulbs were surface-sterilized with clean water, 75% ethanol for 10 s, and sodium hypochlorite for 5 min. Then, the tissues were ground with quartz sand and PBS buffer to obtain a liquid homogenate. PBS buffer was added to each group according to the weight of the group (3 mL/g). The homogenate was diluted 10 times and 100 µL from each sample was plated on solid YPD with 5 mM mannose and streptomycin. The plates were incubated at 28 °C and after 2 days the colonies on each plate were counted. The differences in the number of colonies among different treatment groups were compared via significance analysis.

### 2.12. Statistical Analysis

The experiments were conducted independently three times. GraphPad Prism 8 software was used to plot the curves. The data were analyzed using a one-way ANOVA for multi-comparison analysis.

## 3. Results

### 3.1. Phylogenetic Analysis of Phosphomannose Isomerase in Foc TR4

To identify putative PMI orthologs in *Foc* TR4, a tBlastn search was performed using *S. cerevisiae* PMI (Uniprot No. P29952) as a query, which resulted in the identification of two PMI proteins (PMI1, EXM06237.1 and PMI2, EXL94263.1). The identity and similarity between PMI1 and PMI2 are 35.2% and 50.4%, respectively. The *pmi1* gene in *Foc* TR4 is 1854 bp long and contains two exons and a single intron. The cDNA region is 1347 bp and encodes a protein of 448 amino acids in length (Uniprot No. N1RBD6). The *pmi2* in *Foc* TR4 is 1685 bp long and has only one exon. The cDNA region is 1266 bp and encodes a protein of 421 amino acids in length (Uniprot No. X0IZ75). A constructed phylogenetic tree using protein sequences revealed that *Foc* TR4 PMI1 shares 99.78% identity with *F. oxysporum* f. sp. *lycopersici* 4287 and 33.41% identity with *Neurospora crassa* (Appendix A). However, *Foc* TR4 PMI2 is closer to *F. oxysporum* f. sp. *cubense* race 1, sharing 99.76% identity.

### 3.2. Focpmi1 Exhibits High Expression during Conidial, Germination and Mycelium Stages

The qRT-PCR analysis of *Focpmi1* and *Focpmi2* expression levels at different growth stages of *Foc* TR4 revealed that *Focpmi1* is expressed at all stages of cell growth. At 6 h and 24 h, the expression level of *Focpmi1* was over 2 times and almost 14 times higher, respectively, than that at the 0 h conidia stage (Figure 1A). In contrast, the expression of *Focpmi2* was significantly lower at all growth stages in comparison with the *Focpmi1* expression (Figure 1B). The transcript level of *Focpmi2* was almost negligible during the germination and mycelium growth stages.

### 3.3. Generation of the pmi1 Knockout Mutant and Revertant Strain

To investigate the biological role of PMI1 and PMI2 in *Foc* TR4, a homologous recombination approach was used to create mutant strains (Appendix A). Fusion fragments containing the hygromycin resistance gene *hyg* flanked by 1.1 kb of upstream and downstream regions of the *pmi1* or *pmi2* were transformed into WT protoplasts to replace the *pmi1* or *pmi2* genes, respectively. The Δ*Focpmi1* mutant was selected on regeneration plates containing both hygromycin and mannose. A similar screening approach was applied for the Δ*Focpmi2* mutant, but the obtained transformants grew well in the absence of mannose. Based on phylogenetic tree and qRT-PCR analysis, PMI1 was identified as the key enzyme involved in the conversion between Glc6P and Fru6P, and therefore further characterization was focused on *Focpmi1*. The revertant (RT) strain of Δ*Focpmi1* was constructed by amplifying the *up-pmi1-down* fragment from the WT genomic DNA and transforming it into the mutant protoplasts using mannose as the selection agent. PCR analysis using four primer pairs was performed to confirm the correctness of the strains (Appendix A), and Southern blotting confirmation was conducted using the upstream 1.1 kb region as a probe (Appendix A).

### 3.4. The ∆Focpmi1 Mutant Requires Exogenous Mannose for Growth

To investigate the growth requirements of the Δ*Focpmi1* mutant, we tested the effect of different concentrations of mannose on its growth. Similar to *pmi* deletion strains in other fungal organisms such as *A. fumigatus* [27] and *A. flavus* [26], the Δ*Focpmi1* mutant also required exogenous mannose for growth and was unable to grow in its absence (Figure 2). The mutant showed optimal growth at 5 mM mannose supplementation, while less than 5 mM was insufficient to support growth. On the other hand, higher than 10 mM gradually repressed the growth rate of the mutant. Moreover, more than 30 mM mannose completely abolished growth.

### 3.5. The Growth of the ∆Focpmi1 Mutant Could Not Be Supported by Fructose and Other Carbon Sources

In addition to mannose, other carbon sources such as fructose were added into YPD to test their ability to support the growth of the ∆*Focpmi1* mutant. However, unlike mannose, the mutant was unable to grow in any of the tested concentrations of fructose (Appendix A), suggesting that only mannose can rescue the metabolic pathways affected by PMI1 deficiency. The mutant was also unable to grow on any of the other carbon sources, including Glc, mannose, GlcN, glycerol, xylose, ethanol, arabinose, GlcNAc, sucrose, galactose, and maltose, when used as the sole carbon source in minimal medium (MM), with the exception of fructose (Appendix A). This phenotype is similar to that of the Δ*pmiA* mutant in *A. flavus*, indicating the similar metabolic defects caused by *A. flavus* Δ*pmiA* and *∆Focpmi1*.

### 3.6. Focpmi1 Is Involved in Vegetative Growth but Has No Effect on Conidial Production

To investigate the role of *Focpmi1* in the vegetative growth of *Foc* TR4, we utilized YPD medium supplemented with 5 mM mannose (YPDM) to examine the growth phenotype of the WT, ∆*Focpmi1*, and RT strains. After growing for 5 and 10 days, the mutant strain displayed a lower radial growth rate compared to the WT and RT strains (Figure 3A,B). The growth rate pattern between the WT and RT strains remained consistent. Additionally, the effect of *Focpmi1* deletion on conidial production was not significantly different among the WT, ∆*Focpmi1*, and RT strains (Figure 3C). These findings suggest that *Focpmi1* is involved in vegetative growth but not conidial production.

### 3.7. Focpmi1 Is Involved in Maintaining Cell Wall Integrity and Stress Tolerance

Given that GDP–mannose is a major source of mannosyl units for cell wall glycoproteins and glycoconjugates, we conducted additional tests to investigate the sensitivity of the ∆*Focpmi1* towards cell wall disrupting agents Congo red (CR) and Calcofluor white (CFW). After incubation for 48 h on MM plates supplemented with 5 mM mannose (MMM), the mutant exhibited increased susceptibility to CR and hypersensitivity to CFW as compared to the WT and RT strains (Figure 4A).

However, no significant difference in sensitivity of the mutant was observed towards CR and CFW when grown on YPDM as compared to the WT and RT strains (Appendix A). Analyzing the cell wall components revealed significantly lower chitin levels in the ∆*Focpmi1* strain compared to the WT and RT strains, which is necessary for the rigidity of cell wall (Figure 4B), resulting in the observed hypersensitivity to CFW. Moreover, the mutant exhibited increased sensitivity to H_2_O_2_ on MMM, but not to SDS, sorbitol, and ER stress reagents Brefeldin A and TCEP. In contrast, no significant sensitivity of the mutant was observed towards SDS, H_2_O_2_, sorbitol, Brefeldin A, and TCEP on YPDM compared to the WT and RT strains (Appendix A). These findings suggest that *Focpmi1* plays a vital role in maintaining the cell wall integrity of *Foc* TR4 and that other carbon sources in YPD can serve as alternatives for endogenous mannose.

### 3.8. Focpmi1 Modulates the Expression of Genes Involved in Carbohydrate, Lipid, and Amino Acid Metabolism

To better understand the molecular mechanisms underlying the defective phenotypes observed in the absence of *pmi1*, RNA-seq analysis was conducted to measure global changes in gene expression. Compared to the WT, ∆*Focpmi1* displayed significant up-regulation of 1468 genes and down-regulation of 1190 genes (Fold Change ≥ 5 and FDR < 0.01) (Figure 5A). COG enrichment analysis of the differentially expressed genes (DEGs) showed that carbohydrate, amino acid, and lipid transport and metabolism, as well as secondary metabolism and energy production, were the major pathways affected by the deletion of *Focpmi1*, with each of them having over 100 DEGs (Figure 5B). Interestingly, among the up-regulated genes, 131 genes involved in amino acid transport and metabolism were up-regulated, while 100 genes involved in carbohydrate transport and metabolism were down-regulated (Figure 5C,D). Collectively, the RNA-seq results suggest that the deletion of *Focpmi1* not only disrupted carbohydrate metabolism, but also affected amino acid and lipid metabolism.

### 3.9. Focpmi1 Is Required for Foc TR4 Infection and Virulence

Transcriptomic analysis between the WT and ∆*Focpmi1* revealed that majority of the genes responsible for degrading carbohydrates required for the degradation of plant cell walls and successful invasion of host plants were down-regulated (Appendix A). Furthermore, the mutant exhibited a defective cell wall, which is the outermost organelle that interacts with host plants. Therefore, we hypothesize that the deletion of *Focpmi1* may adversely impact the virulence of *Foc* TR4 when compared to the WT strain.

To test our hypothesis, 2 × 10^6^/mL conidia from the WT, ∆*Focpmi1* and RT strains were applied to infect the Cavendish banana plantlets, as described in materials and methods. After 28 days, the plantlets infected with the Δ*Focpmi1* showed only slight wilt symptoms, with less yellowing of leaves and browning of stems, and maintained normal growth rates. In contrast, the WT and RT infected plantlets exhibited typical wilt symptoms (Figure 6A and Figure 7A). We graded the yellowing of leaves on a scale from 0 (no symptoms) to IV (severe symptoms) (Figure 6B). As shown in Figure 6C, only 17% of the ∆*Focpmi1* infected plants developed yellowing of leaves, whereas 92% of the WT and 75% of the RT infected plantlets displayed disease symptoms of grades III and IV. Furthermore, the growth height of the plants infected with the mutant was not significantly different from that of the non-*Foc* TR4 infected plants, but the heights were significantly lower in the WT and RT infected plants (Figure 6D). These results clearly demonstrate that *Focpmi1* is required for the virulence of *Foc* TR4.

Similarly, the severity of rhizome disease was classified into five grades, with grade 0 indicating no symptoms and grade V indicating severe browning, as illustrated in Figure 7A. As depicted in Figure 7B,C, rhizome browning was absent in plants infected with ∆*Focpmi1*, whereas 75% of WT and 83% of RT-treated plants developed grades III and IV rhizome browning. To assess the fungal burden in rhizomes of CK-, WT-, ∆*Focpmi1-*, and RT-infected banana plantlets, the rhizomes were ground and *Foc* TR4 colonies were recovered (Figure 7D). As compared to the CK group, almost no *Foc* TR4 cells were detected in ∆*Focpmi1*-infected rhizomes, which was significantly different from the WT- and RT-infected rhizomes, where 211 and 185 colonies were detected, respectively (Figure 7E). Collectively, these results provide evidence that *Focpmi1* is necessary for the infection of banana plants.

## 4. Discussion

Bananas are herbaceous plants belonging to the genus Musa and are widely grown in over 100 countries throughout the tropics and subtropics. They produce an estimated 98 million tons of fruit annually, with about one-third of this production coming from Africa, the Asia-Pacific, Latin America, and the Caribbean [36]. Bananas are one of the most important fruit crops in the world, ranking fourth in significance after corn, rice, and wheat [37]. However, this valuable crop is facing serious threats from pathogenic microbes [38,39], particularly the *Fusarium oxysporum* species complex (FOSC), which includes multiple strains that cause vascular diseases in many important crops worldwide [40]. Among the FOSC, the soil-borne filamentous fungal pathogen *F. oxysporum* f. sp. *cubense* is the primary cause of banana fusarium wilt. *Foc* TR4 is the most devastating banana pathogen, capable of infecting almost all banana cultivars, including the widely grown Cavendish [41,42]. Currently, there are no known resistant banana cultivars to *Foc* TR4, and the available fungicides provide only limited protection against it. Therefore, discovering novel antifungal targets and fungicides is an urgent strategy to minimize the loss caused by *Foc* TR4.

The fungal cell wall, which is responsible for maintaining fungal shape and integrity, is the outermost layer of the cell. Being absent in animal cells and structurally different from plants cell wall, the fungal cell wall is considered as an excellent and appealing target for developing novel antifungal drugs [43,44,45]. PMI plays a crucial role in connecting glycolysis to GDP-mannose biosynthesis by facilitating the interconversion of mannose-6-phosphate (Man6P) and fructose-6-phosphate (Fru6P). GDP-mannose is the primary mannosyl donor for cell wall glycoproteins and glycoconjugates [29]. Given PMI’s dual role in energy production and cell wall biogenesis, we hypothesize that PMI deficiency in *Foc* TR4 could lead to defects in infection and pathogenesis.

In this study, we discovered two PMI encoding orthologs in *Foc* TR4, namely *Focpmi1* and *Focpmi2*. The *Focpmi2* deficient mutant did not require mannose supplementation, unlike the *Focpmi1* mutant, indicating that *Focpmi2* is not the crucial gene for the conversion of Fru6P and Man6P. Gene expression analysis using qRT-PCR revealed that the *Focpmi1* gene is expressed at all growth stages, with the highest expression observed during the mycelium stage (Figure 1A). In contrast, *Focpmi2* exhibited negligible expression (Figure 1B). We subsequently focused on the functional role of *Focpmi1*, as its deletion resulted in no growth of *Foc* TR4 in media lacking exogenous mannose. This growth requirement for the Δ*Focpmi1* aligns with previous reports of *pmi* mutants in other fungal pathogens, such as *A. fumigatus* and *A. flavus* [26,27]. Our findings demonstrated that the Δ*Focpmi1* mutant started growth at 2 mM mannose, with optimal growth observed in the presence of 5 mM mannose. However, growth rates declined at concentrations higher than 10 mM (Figure 2). The toxicity of high mannose concentrations on the growth of the mutant is likely due to the honeybee effect [46], caused by intracellular ATP depletion resulting from the useless phosphorylation and dephosphorylation of excess mannose.

Our study revealed that only mannose could initiate the growth of the Δ*Focpmi1*, and a blend of fructose in YPD medium was unable to rescue the mutant, demonstrating that the GDP-Mannose pathway is defective due to PMI1 deletion, and *Focpmi1* is crucial for the survival of *Foc* TR4 (Appendix A). However, when fructose was supplied as the sole carbon source in MM medium, the mutant was able to grow (Appendix A). This finding is consistent with our previously reported *A. flavus* Δ*pmiA* mutant [26], in which the uptake of fructose was accompanied by unavoidable mannose impurities that supported the growth of the mutant. Interestingly, when fructose was added to YPD medium with glucose as the main carbon source, the mutant was unable to grow at all (Appendix A). One possible explanation for this phenotype is the involvement of the carbon catabolite repression pathway [47], which causes the fungus to favor glucose over fructose, leaving no opportunity for the mannose impurity to complement the GDP-Mannose pathway. This further confirms the importance of the GDP-Mannose pathway for *Foc* TR4.

The deletion of *Focpmi1* led to reduced vegetative growth compared to the WT and RT strains, highlighting the predominant functional role of this gene in *Foc* TR4. However, there were no significant differences in conidial production (Figure 3C). The vegetative and sexual stages in fungi may or may not be dependent on each other. In fact, deletion of a particular gene sometimes reduces growth but does not have any impact on conidiation. Our previous work revealed that deletion of *pmi* in *A. flavus* affected both growth and conidiation ability [26]. It is possible that in Δ*Focpmi1*, other genes or environmental factors may compensate for the absence of *pmi* in regulating conidiation. This phenomenon is worth further investigation to understand the molecular mechanisms underlying fungal growth and conidiation. Moreover, this result suggests that PMIs have different functions in different species.

The fungal cell wall is primarily composed of glucan, chitin, and glycoproteins [13], with both chitin and glucan contributing to the strength of the cell wall in fungi [48]. Previous studies on *pmi*-disrupted mutants have shown sensitivity to cell wall disrupting agents such as CR and CFW. Similarly, the Δ*Focpmi1* displayed sensitivity to CR and CFW (Figure 4A), indicating defects in cell wall organization. The analysis of cell wall components revealed a significant decrease in chitin content (Figure 4B), which likely contributed to hypersensitivity to the cell wall-disrupting agent CFW. In addition, the Δ*Focpmi1* exhibited higher levels of cell wall mannan content than the WT and RT strains. The high level of mannan indicates that in the absence of PMI activity, all the supplemented mannose is utilized in the biosynthesis of GDP-man, which is the precursor for cell wall mannan. Furthermore, the mutant showed sensitivity to oxidative stress induced by H_2_O_2_, suggesting that deficiency in *Focpmi1* led to reduced robustness to external stresses.

The differential gene expression patterns observed between the WT and Δ*Focpmi1* suggest a complex regulatory network in response to *Focpmi1* deletion. The up-regulation of genes involved in amino acid, lipid, and carbohydrate metabolism, as well as energy production and conversion, may represent an attempt by the mutant to compensate for the disruption of the GDP-Man pathway. In contrast, the down-regulation of genes involved in carbohydrate transport and metabolism may indicate a prioritization of GDP-Man biosynthesis over other cellular processes. However, the exact regulatory mechanisms underlying these changes in gene expression remain unclear and warrant further investigation. Metabolomics analysis may provide additional insights into the interplay between different metabolic pathways and how they are regulated in response to *Focpmi1* deletion.

PMI mutants in other organisms, such as *A. flavus*, *M. acridum*, *L. mexicana*, and *C. neoformans*, have been found to be avirulent in plant or animal hosts [26,28,29,30]. Similarly, our study shows that the Δ*Focpmi1* mutant is unable to cause disease symptoms in banana plantlets, in contrast to the WT and RT strains, which induced severe wilt symptoms. This was probably due to two reasons: firstly, the Δ*Focpmi1* mutant could not germinate and grow in the nutrient solution, thus it could not initiate the infection; secondly, from the transcriptomic analysis, the decrease in pathogenicity may be related to the down-regulation of many cell wall-degrading enzymes like glucosidases, hydrolases, chitinases, glycosidases, and pectin lyases (Appendix A), which are required by the pathogens for successful penetration into host plants. These findings indicate that *Focpmi1* is necessary for the infection and pathogenesis of *Foc* TR4 and is consistent with the role of *pmi* genes in other fungal pathogens that are important for virulence and pathogenesis.

## 5. Conclusions

In summary, our study demonstrated that deleting the *Focpmi1* gene led to inhibited growth, impaired cell wall integrity, and reduced virulence in *Foc* TR4. Considering that PMIs are either absent or less significant in the Plantae kingdom, our results indicate that targeting *Focpmi1* could be a viable strategy for combating *Foc* TR4 and managing the FWB.

## Figures and Tables

**Figure 1 jof-09-00478-f001:**
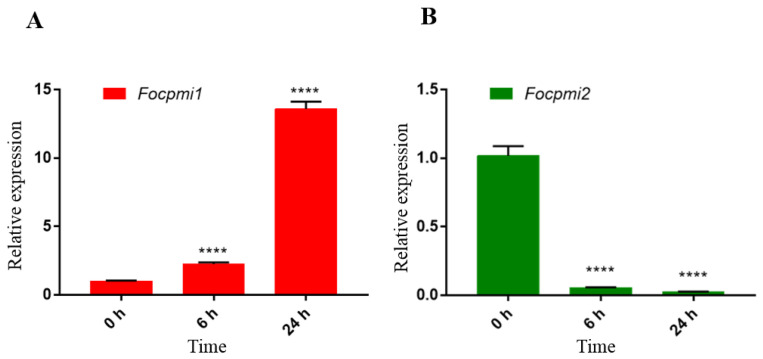
The qRT-PCR expression analysis of *Focpmi1* and *Focpmi2* at conidial (0 h), germination (6 h), and mycelium (24 h) stages cultured in YPD medium. The constitutive *FocEF1α* was used as an internal reference gene and the 0 h stage was used as a control. (**A**) *Focpmi1* expression at different growth stages. (**B**) *Focpmi2* expression at different growth stages. The data are presented as means (±SEM) based on three independent experiments. Asterisks indicate significant differences from the control group (Student’s *t*-test, **** *p* < 0.0001).

**Figure 2 jof-09-00478-f002:**
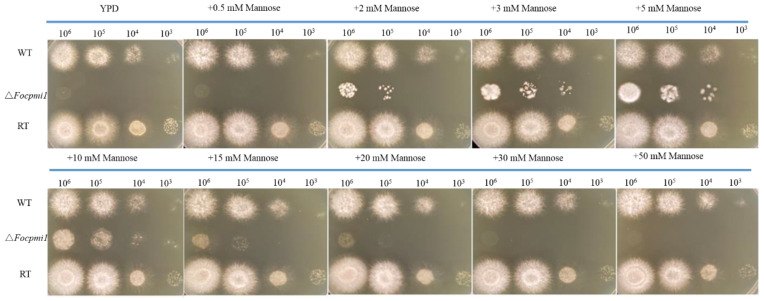
Growth of the WT, ∆*Focpmi1*, and RT strains on YPD supplemented with various concentrations of mannose. Freshly harvested conidia were serially diluted (10^6^–10^3^) and spotted on YPD media supplemented with 0, 0.5, 2, 3, 5, 10, 15, 20, 30, and 50 mM mannose. Plates were incubated at 28 °C for 48 h.

**Figure 3 jof-09-00478-f003:**
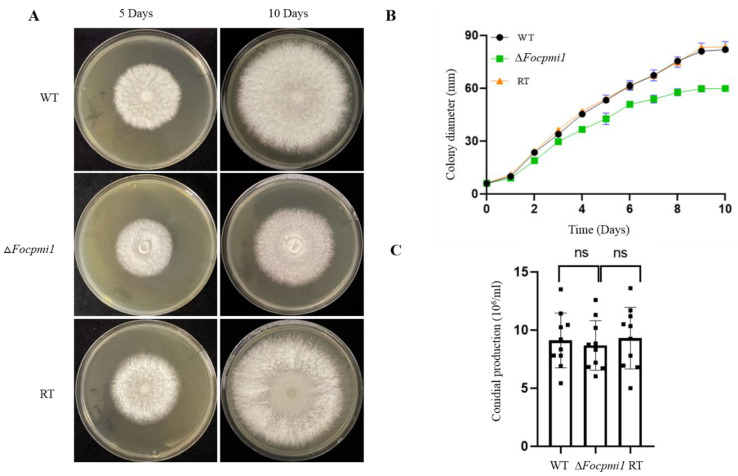
Vegetative growth of the WT, ∆*Focpmi1*, and RT strains on YPD supplemented with 5 mM mannose (YPDM). (**A**) Colony morphology of all the three strains. Freshly harvested conidia of the indicated strains were point inoculated onto YPDM plates and incubated at 28 °C for 5 and 10 days. (**B**) Colony diameter recorded for continuous 10 days after initial inoculation. (**C**) Conidia production of all the three strains. The data represents the mean ± SD from three independent experiments.

**Figure 4 jof-09-00478-f004:**
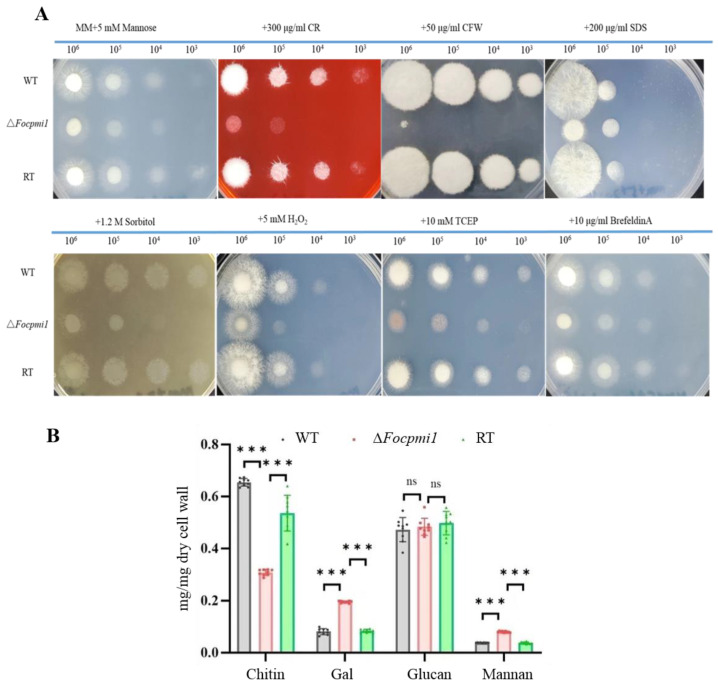
Cell wall integrity, content analysis, and sensitivity of the mutant towards osmotic, oxidative, and ER stresses. (**A**) Freshly harvested serially diluted conidia (10^6^–10^3^) were point inoculated onto MMM plates supplemented with 300 µg/mL CR, 50 µg/mL CFW, 200 µg/mL SDS, 1.2 M sorbitol, 5 mM H_2_O_2_, 10 mM TCEP, and 10 µg/mL Brefeldin A. The plates were photographed after 48 h of incubation at 28 °C. (**B**) A total of 2 × 10^8^ conidia of the WT, ∆*Focpmi1*, and RT strains were inoculated in MMM liquid medium and incubated at 28 °C for 72 h. Dried mycelia (10 mg) were used for the cell wall contents’ quantification. The experiment was conducted in three biological replicates. Asterisks indicate significant differences (Student’s *t*-test, *** *p* < 0.001).

**Figure 5 jof-09-00478-f005:**
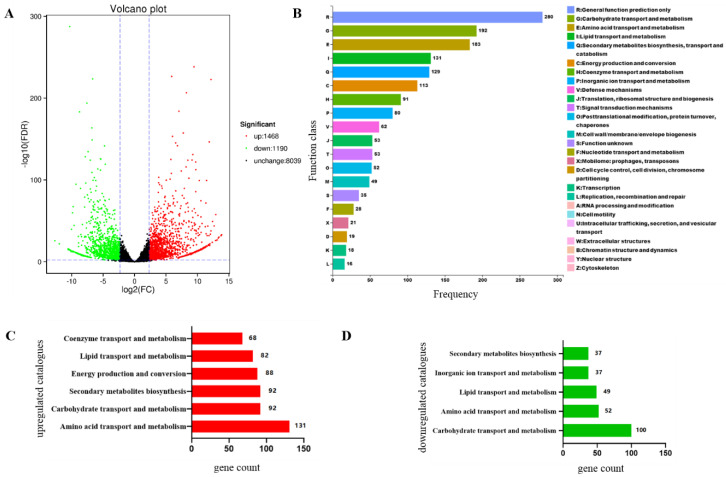
Transcriptomic analysis of the WT and ∆*Focpmi1* mutant. (**A**) Volcano plots displaying the differential gene expression between WT and the mutant. (**B**) COG functional annotation showing the distribution of differentially expressed genes between the two strains. (**C**) Pathway analysis of up-regulated genes and (**D**) pathway analysis of down-regulated genes, revealing the most influenced and disturbed pathways.

**Figure 6 jof-09-00478-f006:**
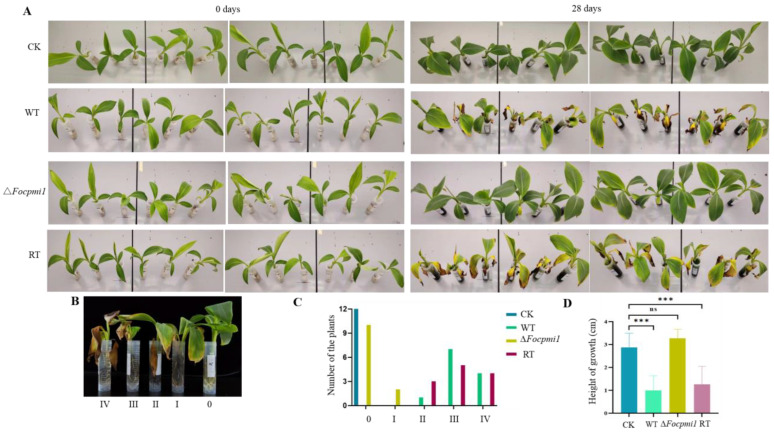
Leaf wilt symptoms from the susceptible banana plants infected by the indicated strains. (**A**) Banana plants inoculated with the WT, ∆*Focpmi1*, and RT strains and CK (no spores) were observed for 28 days to monitor the development of fusarium wilt syndrome. (**B**) The severity of leaf yellowing was graded from 0 (no symptoms) to IV (complete yellowing). (**C**) The number of plants developing leaf disease severity was recorded for each infected group. (**D**) Height of banana plants infected by indicated strains. The data represents the mean ± *SD* from three independent experiments. Asterisks indicate significant differences (Student’s *t*-test, *** *p* < 0.001).

**Figure 7 jof-09-00478-f007:**
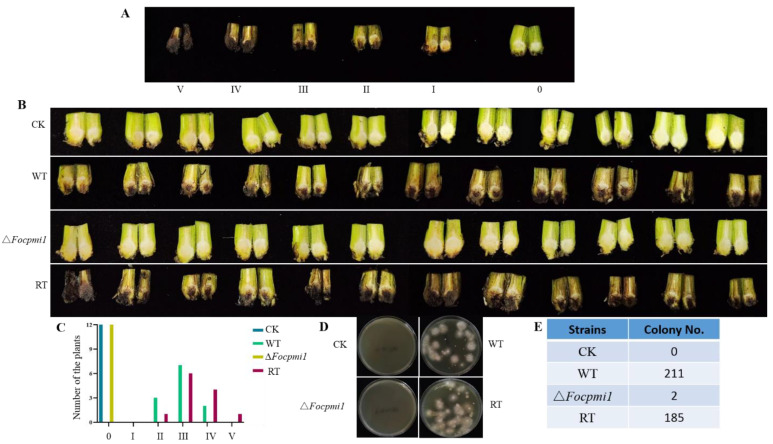
Rhizome wilt symptoms from the susceptible banana plants infected by the indicated strains. (**A**) Grading of rhizome browning from 0 (no symptoms) to V (complete browning). (**B**) The browning degree in banana rhizome tissue after 28 days inoculation with WT, ∆*Focpmi1*, RT, and CK groups. (**C**) Number of the plants developing rhizome disease severity from each infected group. (**D**) Fungal burden recovered from the rhizomes of banana plants infected with each indicated strain. (**E**) Number of fungal colonies recovered after culturing the ground rhizome infected with the indicated strains.

## Data Availability

Data are contained within the article or Appendix A.

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
