# Peer review of "Loss of Phosphomannose Isomerase Impairs Growth, Perturbs Cell Wall Integrity, and Reduces Virulence of Fusarium oxysporum f. sp. cubense on Banana Plants"

_jof, 2023, doi:10.3390/jof9040478_

Round 1

Reviewer 1 Report

The manuscript presented a novel and interesting work however the title is very simple, not concise as well as not reflecting the work done. I suggest the authors modify the title.

 Abstract. Abstract is the most important thing in the research article that must stand alone. In the current abstract, a long and unnecessary background is provided, authors should decrease and provide a maximum of 2-3 lines. The methodology is just one line which needs to be presented in a bit of detail. Results should be presented comprehensively.  

 Authors should revise the manuscript and correct the typos. Like grammatical error, singular and plural terms etc etc. Be consistent while using the italic or non-italic form, should not use mix. Like sometime fusarium in italic while in other place its non-italic. Also please check the other terms or abbreviation as presented in the results figures or table like “Wild type” is presented by WT, while mutant is presented by Δfocpmi1 in figures only . ???? Check such mistakes thoroughly in the manuscript.

 The introduction part is generally good and provided a detailed work done in the field. However, the objectives should be clearly mentioned at the end of the introduction rather than discussing the results or suggestions in this section.

 Methodology Part

Provide the reference (s) for the procedure of the phylogenetic tree.

 In section 2.4, the authors mention that the hygromycin resistance marker gene was amplified using primer pair P3. However, there is no information provided on where this marker gene was amplified from. It would be helpful to include this information in the paper to provide more detail on the construction of the deletion mutant and revertant strain.

Provide the reference(s) for the pathogenicity assay.

 Results Part

It will be interesting to present Figure S1, in the main manuscript rather than in supplementary file.

 4) Figure S4 shows that the ∆Focpmi1 mutant was unable to grow on a variety of different carbon sources in minimal medium, with the exception of fructose. However, the mutant was also unable to grow even when mannose was present. How? Why the mutant was not able to grow when mannose was present, despite being able to grow when fructose was present.

Therefore, authors should provide a solid reason and explain in the text and in figure caption.

 Figure S6, describes “Cell wall content analysis of ∆Focpmi1 strain” which is the core finding of the present study, therefore, I suggest presenting this in the main manuscript to reflect the significant of the study and ease for the readers in the field.

 Discussion

The discussion part needs a modification to clearly discuss the results. The loss of pathogenicity of the mutant which, according to transcriptomic analysis, is because of the down regulation of genes related to carbohydrate metabolism. The authors also need to mention this down regulation in the discussion section.

Author Response

Point-by-point response:

Reviewer 1:

The manuscript presented a novel and interesting work however the title is very simple, not concise as well as not reflecting the work done. I suggest the authors modify the title.

Thanks for the suggestion. We have modified the title.

 Abstract. Abstract is the most important thing in the research article that must stand alone. In the current abstract, a long and unnecessary background is provided, authors should decrease and provide a maximum of 2-3 lines. The methodology is just one line which needs to be presented in a bit of detail. Results should be presented comprehensively.

Thanks for the comment. We have revised the abstract, and presented in a better way in the revised manuscript.

 Authors should revise the manuscript and correct the typos. Like grammatical error, singular and plural terms etc etc. Be consistent while using the italic or non-italic form, should not use mix. Like sometime fusarium in italic while in other place its non-italic. Also please check the other terms or abbreviation as presented in the results figures or table like “Wild type” is presented by WT, while mutant is presented by Δfocpmi1 in figures only . ???? Check such mistakes thoroughly in the manuscript.

Thanks for the comment. We have revised the manuscript and corrected all the mentioned mistakes.

 The introduction part is generally good and provided a detailed work done in the field. However, the objectives should be clearly mentioned at the end of the introduction rather than discussing the results or suggestions in this section.

Thanks for the comment. We have added specific objectives at the end of introduction.

 Methodology Part

Provide the reference (s) for the procedure of the phylogenetic tree.

Thanks for pointing this out. We have added the reference.

 In section 2.4, the authors mention that the hygromycin resistance marker gene was amplified using primer pair P3. However, there is no information provided on where this marker gene was amplified from. It would be helpful to include this information in the paper to provide more detail on the construction of the deletion mutant and revertant strain.

Thanks for the comment. We have added the amplification source of hygromycin marker in the revised version.

Provide the reference(s) for the pathogenicity assay.

Thanks for the comment. We have added the reference for the pathogenicity assay in the revised version.

 Results Part

It will be interesting to present Figure S1, in the main manuscript rather than in supplementary file.

Thanks for the comment. We don’t think that it will be better to shift the Fig. S1 to the main manuscript. However, if the reviewer insists, we will shift.

 4) Figure S4 shows that the ∆Focpmi1 mutant was unable to grow on a variety of different carbon sources in minimal medium, with the exception of fructose. However, the mutant was also unable to grow even when mannose was present. How? Why the mutant was not able to grow when mannose was present, despite being able to grow when fructose was present. Therefore, authors should provide a solid reason and explain in the text and in figure caption.

Thanks for the comment. In Fig. S4, the mutant is unable to grow in the presence of mannose when it is the sole carbon source (clearly mentioned in the figure legend), however the mutant is able to grow in the presence of fructose as the sole carbon source. This is because mannose is speculated to be an unavoidable impurity during fructose production, and although fructose is the sole carbon source, a small amount of mannose is present to support growth. These details are clearly discussed in the discussion section of the manuscript.

 Figure S6, describes “Cell wall content analysis of ∆Focpmi1 strain” which is the core finding of the present study, therefore, I suggest presenting this in the main manuscript to reflect the significant of the study and ease for the readers in the field.

Thanks for the suggestion. Surely this is the main result, and we have shifted to the main manuscript in the revised version.

 Discussion

The discussion part needs a modification to clearly discuss the results. The loss of pathogenicity of the mutant which, according to transcriptomic analysis, is because of the down regulation of genes related to carbohydrate metabolism. The authors also need to mention this down regulation in the discussion section.

Thanks for the suggestion. We have revised and added the discussion about loss of pathogenicity and genes involved in it.

Reviewer 2:

The authors presented a very polished and well-written manuscript detailing the role of phosphomannose isomerase (PMI) in the growth, cell wall integrity and pathogenicity of F. oxysporum f.sp. cubense, affecting banana cultivars. The authors performed a thorough series of experiments that support their hypothesis. They first performed sequence comparison of S. cerevisiae PMI to obtain PMI orthologs in F. oxysporum f.sp. cubense. After detecting two candidate orthologs, they used qRT-PCR to determine the expression levels of both candidates, showing that one had increased expression throughout the growth stages while the other did not, suggesting that the latter was not functional. The authors then developed knockout mutants and tested the different factors (growth in different media, cell wall integrity analysis, and transcriptome analyses between the mutant and wild type). These series of experiments demonstrated the effects of PMI and mention that this enzyme could be a viable strategy for combatting F. oxysporum f. sp. cubense.

I would like to commend the authors for such detailed work and their presentation of results was outstanding. I  did not detect any grammatical errors or unclear descriptions, with the exception of the following typos:

1) bracket on page 3, paragraph "[19-21" should read "[19-21]".

Thanks for the comment. We have revised.

2) Fig 5. C and D. Title of y axis is mispelled. Should read "upregulated", not "upreguated" and "downregulated", not "downreguated", respectively.

Thanks for the comment. We have made corrections.

3) Reference 14 is not mentioned in the text

Thanks for pointing out. We have made corrections in the revised version.

  • Table S1 could be single-spaced, to shorten its length

Thanks for the comment. We have revised.

Reviewer 3:

Abstract : Need to be improved. Below are some specific comments.

  • In the first sentence, replace ‘pathogen’ with ‘pathogens’.

Thanks for the comment. We have modified the abstract in the revised version.

  • The specific results may be given to support the statements. For instance, some significant results about hypersensitivity to cell wall and oxidative stress.

Thanks for the comment. We have revised the overall abstract and added the above information.

  • Give some examples of up and down regulated genes in transcriptomic analysis.

Thanks for comment. In the revised manuscript we have added some of the classes of genes which are necessary in the degrading plant cell wall from the transcriptomic data in the discussion section.

  • What is meant by low colonization and attenuated virulence? Some evidence need to be mentioned.

Thanks for the comment. In the revised abstract version we have added that the mutant exhibited reduced virulence in banana plants.

  • I would suggest to remove or shorten the 12 initial lines of the abstract that are focused on introduction. And add more about the study done here. Add some significant results and concrete evidences to support the claims in the abstract.

Thanks for the suggestions. We have revised the abstract in the revised manuscript, and covered all the suggestions.

Introduction:

Introduction is well written and comprehensive. It has been explained that similar studies have been recently published in other pathogenic fungi such as Aspergillus flavus by the same author. The author therefore expects similar results in Foc. There are some minor issues about English writing e.g., sentences should not be started with abbreviations.

Thanks for the comments. We have thoroughly revised and made corrections in the revised manuscript.

Materials and Methods and Results are well explained and nicely written.

 Discussion: Well written. However, I would have a few questions as follows:

  • “One possible explanation for this phenotype is the involvement of the carbon catabolite repression pathway”. To me it doesn’t make sense. There may be some other explanation to this phenotype?

Thanks for the comment. Based on our observations, we believe that this is the most plausible explanation for the observed phenotype. As shown in Fig. S3, the addition of fructose to YPD medium is unable to restore the mutant growth. This is because the mutant prefers glucose as its primary carbon source, leading to glucose uptake while leaving fructose uptake unused. However, when fructose is the only available carbon source, as shown in Fig. S4, the mutant is able to utilize fructose as its carbon source, resulting in the expected presence of mannose as an impurity, which complements growth. This explanation is already mentioned in the discussion section.

  • “However, there were no significant differences in conidial production (Fig. 3C)”. Here I would comment if there is a role in growth, how it is possible to not have any role on conidiation? Or conidiation has no relation with growth? Need a little more discussion.

Thanks for the comment. The vegetative and sexual stages in fungi may or may not affect each other, and may be independent of each other. In fact, deletion of a particular gene sometimes reduces growth and affects its conidiation ability. Our previous work revealed that deletion of pmi in Aspergillus flavus affected both growth and conidiation ability. However, in Foc TR4, pmi absence only affected growth and not conidiation. It is possible that other genes or environmental factors may compensate for the absence of pmi in regulating conidiation. This phenomenon is worth further investigation to understand the molecular mechanisms underlying fungal growth and conidiation. We have added these details in the discussion section.

  • “Similarly, our study shows that the ΔFocpmi1 mutant is unable to cause disease symptoms in banana plantlets, in contrast to the WT and RT strains which induced severe wilt symptoms.” Here, the lack of ability to cause infection by the mutants: is it due to the inability to survive in the environment? May be if you supply mannose in the root environment, they also cause infection? Need more discussion.

Thanks for the comment and raising the question. Actually, before performing the virulence assay, we tested the mutant in the nutrient solution with various concentrations of mannose, and found that the mutant could not germinate due to the sucrose component in the nutrient solution blocking the uptake of mannose. Thus, we speculate that the reduced virulence and pathogenesis of mutant were probably due to two reasons: firstly, the ΔFocpmi1 mutant could not germinate and growth in the nutrient solution, thus it could not initiate the infection; secondly, from the transcriptomic analysis, the decrease in pathogenicity may be related to the down regulation of many cell wall degrading enzymes like glucosidases, hydrolases, chitinases, glycosidases, and pectin lyases (Table S2), which are required by the pathogens for successful penetration into host plants. We have added this explanation in the revised manuscript.

Reviewer 4

The manuscript show interesting results about the role of Focpmi1 and mannose in the growing of Fusarium oxysporum f. sp. cubense. I just suggest some minor corrections:

 Minor corrections:

- In Abstract, change "involved in physiological processes ." to "involved in physiological processes." (exclude the space between "processes" and the end point).

Thanks for pointing out. We have revised the whole abstract.

- In Introduction, put references in the sentence: "The functions of PMI have also been reported in several other pathogenic organisms, including Cryptococcus neoformans, Leishmania mexicana, and Metharizium acridum."

Thanks for pointing out. We have added the above references in the revised manuscript.

- In discussion, format "Fusarium oxysporum" in italic style, in the sentence "Fusarium oxysporum species complex (FOSC), which includes multiple strains that cause vascular diseases in many important crops worldwide."

Thanks for the comment. We have made the correction in the revised manuscript.

- In Supplementary Material, format "cubense", "vasinfectum", "lycopersici", "rapae", and "graminearum" in italic style (Subtitle of Fig. S1).

Thanks for the comment. We have made corrections in the revised version.

- In Supplementary Material, put the caption of Table S2 above the table and not below it. 

Thanks for the comment. We have corrected that in the revised version. 

Reviewer 2 Report

The authors presented a very polished and well-written manuscript detailing the role of phosphomannose isomerase (PMI) in the growth, cell wall integrity and pathogenicity of F. oxysporum f.sp. cubense, affecting banana cultivars. The authors performed a thorough series of experiments that support their hypothesis. They first performed sequence comparison of S. cerevisiae PMI to obtain PMI orthologs in F. oxysporum f.sp. cubense. After detecting two candidate orthologs, they used qRT-PCR to determine the expression levels of both candidates, showing that one had increased expression throughout the growth stages while the other did not, suggesting that the latter was not functional. The authors then developed knockout mutants and tested the different factors (growth in different media, cell wall integrity analysis, and transcriptome analyses between the mutant and wild type). These series of experiments demonstrated the effects of PMI and mention that this enzyme could be a viable strategy for combatting F. oxysporum f. sp. cubense. 

I would like to commend the authors for such detailed work and their presentation of results was outstanding. I  did not detect any grammatical errors or unclear descriptions, with the exception of the following typos:

1) bracket on page 3, paragraph "[19-21" should read "[19-21]". 

2) Fig 5. C and D. Title of y axis is mispelled. Should read "upregulated", not "upreguated" and "downregulated", not "downreguated", respectively.

3) Reference 14 is not mentioned in the text

4) Table S1 could be single-spaced, to shorten its length

Author Response

(The authors gave the same response as above.)

Reviewer 3 Report

Abstract : Need to be improved. Below are some specific comments. 

·      In the first sentence, replace ‘pathogen’ with ‘pathogens’.

·      The specific results may be given to support the statements. For instance, some significant results about hypersensitivity to cell wall and oxidative stress.

·      Give some examples of up and down regulated genes in transcriptomic analysis.

·      What is meant by low colonization and attenuated virulence? Some evidence need to be mentioned.

·      I would suggest to remove or shorten the 12 initial lines of the abstract that are focused on introduction. And add more about the study done here. Add some significant results and concrete evidences to support the claims in the abstract.

Introduction: 

Introduction is well written and comprehensive. It has been explained that similar studies have been recently published in other pathogenic fungi such as Aspergillus flavus by the same author. The author therefore expects similar results in Foc. There are some minor issues about English writing e.g., sentences should not be started with abbreviations.

Materials and Methods and Results are well explained and nicely written.

Discussion: Well written. However, I would have a few questions as follows:

·     “One possible explanation for this phenotype is the involvement of the carbon catabolite repression pathway”. To me it doesn’t make sense. There may be some other explanation to this phenotype?

·      “However, there were no significant differences in conidial production (Fig. 3C)”. Here I would comment if there is a role in growth, how it is possible to not have any role on conidiation? Or conidiation has no relation with growth? Need a little more discussion.

·      “Similarly, our study shows that the ΔFocpmi1 mutant is unable to cause disease symptoms in banana plantlets, in contrast to the WT and RT strains which induced severe wilt symptoms.” Here, the lack of ability to cause infection by the mutants: is it due to the inability to survive in the environment? May be if you supply mannose in the root environment, they also cause infection? Need more discussion.

Author Response

Reviewer 3:

Abstract : Need to be improved. Below are some specific comments.

  • In the first sentence, replace ‘pathogen’ with ‘pathogens’.

Thanks for the comment. We have modified the abstract in the revised version.

  • The specific results may be given to support the statements. For instance, some significant results about hypersensitivity to cell wall and oxidative stress.

Thanks for the comment. We have revised the overall abstract and added the above information.

  • Give some examples of up and down regulated genes in transcriptomic analysis.

Thanks for comment. In the revised manuscript we have added some of the classes of genes which are necessary in the degrading plant cell wall from the transcriptomic data in the discussion section.

  • What is meant by low colonization and attenuated virulence? Some evidence need to be mentioned.

Thanks for the comment. In the revised abstract version we have added that the mutant exhibited reduced virulence in banana plants.

  • I would suggest to remove or shorten the 12 initial lines of the abstract that are focused on introduction. And add more about the study done here. Add some significant results and concrete evidences to support the claims in the abstract.

Thanks for the suggestions. We have revised the abstract in the revised manuscript, and covered all the suggestions.

Introduction:

Introduction is well written and comprehensive. It has been explained that similar studies have been recently published in other pathogenic fungi such as Aspergillus flavus by the same author. The author therefore expects similar results in Foc. There are some minor issues about English writing e.g., sentences should not be started with abbreviations.

Thanks for the comments. We have thoroughly revised and made corrections in the revised manuscript.

Materials and Methods and Results are well explained and nicely written.

 Discussion: Well written. However, I would have a few questions as follows:

  • “One possible explanation for this phenotype is the involvement of the carbon catabolite repression pathway”. To me it doesn’t make sense. There may be some other explanation to this phenotype?

Thanks for the comment. Based on our observations, we believe that this is the most plausible explanation for the observed phenotype. As shown in Fig. S3, the addition of fructose to YPD medium is unable to restore the mutant growth. This is because the mutant prefers glucose as its primary carbon source, leading to glucose uptake while leaving fructose uptake unused. However, when fructose is the only available carbon source, as shown in Fig. S4, the mutant is able to utilize fructose as its carbon source, resulting in the expected presence of mannose as an impurity, which complements growth. This explanation is already mentioned in the discussion section.

  • “However, there were no significant differences in conidial production (Fig. 3C)”. Here I would comment if there is a role in growth, how it is possible to not have any role on conidiation? Or conidiation has no relation with growth? Need a little more discussion.

Thanks for the comment. The vegetative and sexual stages in fungi may or may not affect each other, and may be independent of each other. In fact, deletion of a particular gene sometimes reduces growth and affects its conidiation ability. Our previous work revealed that deletion of pmi in Aspergillus flavus affected both growth and conidiation ability. However, in Foc TR4, pmi absence only affected growth and not conidiation. It is possible that other genes or environmental factors may compensate for the absence of pmi in regulating conidiation. This phenomenon is worth further investigation to understand the molecular mechanisms underlying fungal growth and conidiation. We have added these details in the discussion section.

  • “Similarly, our study shows that the ΔFocpmi1 mutant is unable to cause disease symptoms in banana plantlets, in contrast to the WT and RT strains which induced severe wilt symptoms.” Here, the lack of ability to cause infection by the mutants: is it due to the inability to survive in the environment? May be if you supply mannose in the root environment, they also cause infection? Need more discussion.

Thanks for the comment and raising the question. Actually, before performing the virulence assay, we tested the mutant in the nutrient solution with various concentrations of mannose, and found that the mutant could not germinate due to the sucrose component in the nutrient solution blocking the uptake of mannose. Thus, we speculate that the reduced virulence and pathogenesis of mutant were probably due to two reasons: firstly, the ΔFocpmi1 mutant could not germinate and growth in the nutrient solution, thus it could not initiate the infection; secondly, from the transcriptomic analysis, the decrease in pathogenicity may be related to the down regulation of many cell wall degrading enzymes like glucosidases, hydrolases, chitinases, glycosidases, and pectin lyases (Table S2), which are required by the pathogens for successful penetration into host plants. We have added this explanation in the revised manuscript.

Reviewer 4 Report

The manuscript show interesting results about the role of Focpmi1 and mannose in the growing of Fusarium oxysporum f. sp. cubense. I just suggest some minor corrections:

Minor corrections:

- In Abstract, change "involved in physiological processes ." to "involved in physiological processes." (exclude the space between "processes" and the end point).

- In Introduction, put references in the sentence: "The functions of PMI have also been reported in several other pathogenic organisms, including Cryptococcus neoformans, Leishmania mexicana, and Metharizium acridum."

- In discussion, format "Fusarium oxysporum" in italic style, in the sentence "Fusarium oxysporum species complex (FOSC), which includes multiple strains that cause vascular diseases in many important crops worldwide."

- In Supplementary Material, format "cubense", "vasinfectum", "lycopersici", "rapae", and "graminearum" in italic style (Subtitle of Fig. S1).

- In Supplementary Material, put the caption of Table S2 above the table and not below it.  

Author Response

The manuscript show interesting results about the role of Focpmi1 and mannose in the growing of Fusarium oxysporum f. sp. cubense. I just suggest some minor corrections:

 Minor corrections:

- In Abstract, change "involved in physiological processes ." to "involved in physiological processes." (exclude the space between "processes" and the end point).

Thanks for pointing out. We have revised the whole abstract.

- In Introduction, put references in the sentence: "The functions of PMI have also been reported in several other pathogenic organisms, including Cryptococcus neoformans, Leishmania mexicana, and Metharizium acridum."

Thanks for pointing out. We have added the above references in the revised manuscript.

- In discussion, format "Fusarium oxysporum" in italic style, in the sentence "Fusarium oxysporum species complex (FOSC), which includes multiple strains that cause vascular diseases in many important crops worldwide."

Thanks for the comment. We have made the correction in the revised manuscript.

- In Supplementary Material, format "cubense", "vasinfectum", "lycopersici", "rapae", and "graminearum" in italic style (Subtitle of Fig. S1).

Thanks for the comment. We have made corrections in the revised version.

- In Supplementary Material, put the caption of Table S2 above the table and not below it. 

Thanks for the comment. We have corrected that in the revised version.